# Association of early discharge and clinical outcomes following proctectomy for patients with rectal cancer: A NRD analysis

Aboubacar Cherif[1], Troy N. Coaston[1], Melissa Justo[1], Dariush Yalzadeh[1], Saad Mallick[1], Zihan Gao[1], Esteban Aguayo[1], Peyman Benharash[1], Hanjoo Lee[2]*

**1** Cardiovascular Outcomes Research Laboratories (CORELAB), David Geffen School of Medicine, University of California at Los Angeles, Los Angeles, California, United States of America, **2** Division of Colon and Rectal Surgery, Department of Surgery, Harbor-UCLA Medical Center, Torrance, California, United States of America

* H.Lee@dhs.lacounty.gov

## Abstract

### Background

Adoption of enhanced recovery pathways in proctectomy has gained attention due to its potential to reduce resource utilization and optimize recovery. Using a nationally representative cohort, we examined the association between early discharge (within 3 days) and 30-day readmissions, perioperative complications, and hospitalization costs.

### Methods

Adult (≥18 years) patients with rectal cancer undergoing proctectomy were identified using the 2016–2021 Nationwide Readmissions Database. Patients were classified as *Early* if they were discharged within 3 days of proctectomy and otherwise were classified as *Routine.* Multivariable regression models were developed to assess associations between early discharge and 30-day outcomes, as well as associated readmission, costs, and mortality. Entropy balancing was employed to obtain a weighted comparison to adjust for intergroup differences.

### Results

Of an estimated 39,505 patients, 25.8% were discharged early. *Early* was younger (59 [51−68] vs 62 years [53−71], p < 0.001), had lower comorbidity burden (2 [1 −3] vs. 3 [2 −4] unit, p < 0.001), and more frequently received laparoscopic proctectomy (60.9 vs. 39.0%, p < 0.001). Following entropy balancing, early discharge was associated with lower readmission rates (adjusted odds ratio [AOR], 0.77; 95% confidence interval [CI], 0.69–0.86) and decreased cumulative hospitalization costs (β: − $6.2K;

**Data availability statement:** The data utilized in this study is owned by a third-party organization and were derived from the Nationwide Readmissions Database (NRD), which is publicly available upon purchase through the Healthcare Cost and Utilization Project (HCUP). Due to the deidentified nature of the NRD, individual patient data cannot be shared directly by the authors. Access to the NRD is available to eligible researchers through HCUP following data use agreement protocols. HIPAA-compliant, deidentified data can be requested by any researcher at: https://hcup-us.ahrq.gov/tech_assist/centdist.jsp.

**Funding:** The author(s) received no specific funding for this work.

**Competing interests:** The authors have declared that no competing interests exist.

95% CI, −8.6k to −3.7k). Mortality at readmission was similar between the two groups (AOR, 0.69; 95% CI, 0.33–1.45).

## Conclusions

Early discharge after proctectomy is feasible for rectal cancer patients when clinically appropriate and is associated with reduced readmissions and hospitalization costs without compromising patient safety compared to routine. Broader implementation of early discharge protocols may optimize outcomes for rectal cancer patients undergoing proctectomy.

---

## Introduction

There are over 45,000 new cases of rectal cancer diagnosed in the United States annually, and the incidence of rectal cancer in young adults is expected to more than double over the next decade [1]. Proctectomy, specifically total mesorectal excision, is the gold standard for curative resection for rectal cancer. However, proctectomy remains associated with significant morbidity, length of stay and resource utilization [2,3]. To address the anticipated increase in rectal cancer treatments, improving clinical outcomes and reducing the financial costs of proctectomy could help alleviate the growing healthcare burden associated with the disease.

Most recently, enhanced recovery protocols after surgery, or ERAS, have aimed to hasten postoperative recovery after major surgery by modifying perioperative management strategies [4]. Institutional implementation of colorectal ERAS guidelines has led to a notable increase in early discharges, defined as leaving the hospital within 3 days of surgery [5–7]. Although proctectomy is generally more complex than colectomy, reported median lengths of stay for proctectomy (5–6 days) overlap with the 4–7 days commonly reported for colectomy, supporting the use of a similar 3-day threshold to identify patients discharged substantially earlier than contemporary practice patterns [8]. In turn, earlier discharge results in reduced physiological and financial burden on the patient, with an added benefit of decreasing hospital costs [9]. However, most published studies evaluating colorectal ERAS guidelines do not distinguish between rectal and colonic procedures. Although few retrospective studies have shown improved outcomes in proctectomies following implementation of ERAS guidelines [10–12], large-scale studies examining the clinical outcomes of early discharge in patients undergoing proctectomy remain scarce.

To address this, the present study utilized a national database to examine the association of early discharge with clinical outcomes and financial costs following proctectomy for rectal cancer. We hypothesized that early discharge would be linked with decreased adverse outcomes as well as readmission rates and resource utilization.

## Methods

This was a retrospective study of the 2016–2021 Nationwide Readmissions Database (NRD). Maintained by the Health-care Cost and Utilization Project, the NRD is the largest, publicly available all-payer readmissions database and samples discharges from 30 geographically diverse states [13]. The NRD provides accurate estimates for approximately 60% of hospitalizations in the United States through the utilization of robust survey weighting methodology. Unique linkage numbers are used to track readmissions within each state and calendar year.

All elective adult (≥18 years) hospitalizations entailing proctectomy were identified using previously reported International Classification of Diseases-10th Revision (ICD-10) procedure codes (ICD-10-PCS: ODTP0ZZ ODTP4ZZ) [14]. Only patients with rectal cancer, identified using ICD-10 diagnosis code ICD-10-CM: C20, were included in the study. Metastatic disease was similarly identified using ICD-10-CM: C77, C78, C79, C80 and modeled as a composite variable to reflect the overall burden of advanced malignancy. Patients admitted in December of each year were excluded to allow for 30 days of follow-up after discharge. Similarly, patients with a diagnosis of Ulcerative Colitis or Crohn's disease were not included for analysis. Records with missing key variables such as age, sex, and mortality were excluded. Additionally, those who did not have rectal tumor or who died during index hospitalization were not included for further study. Finally, patients with a length of stay greater than 9 days (90th percentile for length of stay) were excluded to decrease the heterogeneity of the cohort.

### Variable definitions and study outcomes

Patient characteristics including age, sex, and insurance coverage were defined in accordance with the NRD Data Dictionary [15]. Facility characteristics such as bed size and teaching status were similarly defined. Institutional operative volume tertiles (low, medium, high) were calculated from proctectomy caseload during each calendar year. The Van Walraven modification of the Elixhauser Comorbidity Index, a validated composite of 30 comorbidities, was used to quantify the burden of chronic conditions as previously described [16]. Individual comorbidities such as congestive heart failure, coronary artery disease, rectal tumor, and metastatic cancer were tabulated using ICD-10 diagnosis codes [17]. Similarly, acute events were ascertained and grouped into cardiac, infectious, intraoperative, respiratory, stroke, and thromboembolic [18]. Based on timing of discharge following proctectomy, patients were categorized into 2 groups: *Early* (within 3 days) and *Routine* (Others). Inpatient costs were generated via application of hospital specific cost-to-charge ratios and inflation adjusted to the 2021 Personal Health Index [19]. Cumulative length of stay (LOS) variable was constructed by summing the index hospitalization LOS with any subsequent readmission occurring within 30 days of discharge. For patient with multiple readmissions, we summed the LOS of the index admission, the first readmission, and the LOS of a second readmission only when the second readmission also fell within the 30 day window.

### Outcomes

The primary study endpoint was readmission within 30 days of initial discharge. Secondary endpoints included development of perioperative complications (cardiac, respiratory, infectious), mortality during readmission, and 30-day hospitalization costs.

### Statistical analysis and cohort definition

Categorical variables are reported as proportions and compared across groups using the Pearson chi-square test. Continuous variables are summarized as medians with interquartile range (IQR) and analyzed with the Man-Whitney *U* test. Subsequently, multivariable logistic regression models were used to assess factors independently associated with in-hospital mortality and perioperative complications, adjusting for patient and hospital characteristics including: patient demographics, comorbidities, insurance type, hospital volume, and surgical approach. Linear regression models were

similarly used for evaluating associations with hospitalization costs and hospitalization length of stay (LOS). To account for inter-group differences, entropy balancing was employed to obtain a weighted comparison group with similar covariate distributions. Standard mean differences (SMDs) were obtained to demonstrate effect size with SMD > 0.1 considered significant. Model covariates were selected using elastic net regularization. Elastic Net adopts a penalized least squares methodology to select model covariates while minimizing overfitting and collinearity via Akaike and Bayesian information criteria [20]. Models were also evaluated using receiver operating characteristics (C-statistic) and calibration plots, as appropriate. Regression outputs are reported as adjusted OR (AOR) or beta coefficients (β) for logistic or linear models, respectively, both with 95% CI.

All statistical analysis was performed using Stata 16.1 software (StataCorp, LLC, College Station, TX). The study was deemed exempt from full review by the Institutional Review Board at the University of California, Los Angeles.

## Results

### Demographic comparison of early discharge and routine discharge

Of 39,505 patients with rectal cancer meeting study requirements, 25.8% were discharged within 3 days of surgery and therefore labeled *Early* (Fig 1). Relative to *Routine*, *Early* was younger (59 [51–68] vs 62 years [53–71], p < 0.001) privately insured (55.0 vs 43.3%, p < 0.001) and more frequently treated at high volume centers (75.7 vs 72.4%, p < 0.001). Furthermore, *Early* had lower prevalence of coronary artery disease (7.8 vs 9.9%, p < 0.001), congestive heart failure (2.1 vs. 4.0%, p < 0.001), and metastatic cancer (17.4 vs 22.7%, p < 00.1), compared to *Routine* (Table 1). Finally, *Early* was more likely to undergo a laparoscopic proctectomy (60.9 vs 39.0%, p < 0.001) and an Ileostomy creation (43.7 vs 37.8%, p < 0.001), but less likely to undergo robotic assistance (14.2 vs 25.7%, p < 0.001) and less likely to have a history of radiation (33.3 vs 39.9%, p < 0.001).

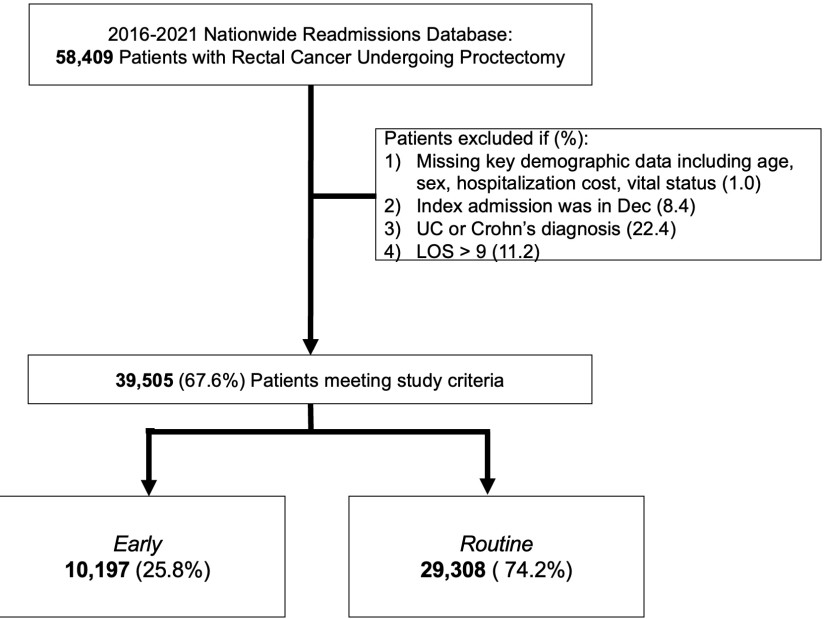

**Fig 1. Study CONSORT diagram.** *Early:* Patients discharged within 3 days of proctectomy; *Routine:* Patients discharged after 3 days of proctectomy. UC: Ulcerative Colitis.

**Table 1. Comparison of baseline patient and hospital characteristics between early discharge and routine discharge patients.**

| Characteristics | *Early* (n = 10,197) | *Routine* (n = 29,308) | p-value |
|---|---|---|---|
| Age (years, median, [IQR]) | 59 [51-68] | 62 [53-71] | <0.001 |
| Elixhauser Index (units, median, [IQR]) | 2 [1 –3] | 3 [2 –4] | <0.001 |
| Female (%) | 40.7 | 40.0 | 0.4 |
| Ileostomy Creation | 43.7 | 37.8 | <0.001 |
| *Income Quartile (%)* | | | <0.001 |
| 76th-100th | 25.5 | 22.0 | |
| 51st-75th | 27.0 | 26.0 | |
| 26th-50th | 26.0 | 27.7 | |
| 0-25th | 21.5 | 24.3 | |
| *Hospital Volume (%)* | | | <0.001 |
| Large | 75.7 | 72.4 | |
| Medium | 16.3 | 16.7 | |
| Small | 8.0 | 11.0 | |
| *Primary Payer (%)* | | | <0.001 |
| Private | 55.0 | 43.3 | |
| Medicare | 32.5 | 41.4 | |
| Medicaid | 8.2 | 11.1 | |
| Uninsured | 1.7 | 1.5 | |
| Other | 2.7 | 2.6 | |
| *Hospital Teaching Status (%)* | | | 0.01 |
| Non-Metropolitan | 2.2 | 3.4 | |
| Metropolitan non-teaching | 12.6 | 13.1 | |
| Metropolitan teaching | 85.2 | 83.5 | |
| *Operative Approach (%)* | | | |
| Open | 39.2 | 61.1 | <0.001 |
| Laparoscopic | 60.9 | 39.0 | <0.001 |
| Robotic Assistance | 14.2 | 25.7 | <0.001 |
| *Comorbidities (%)* | | | |
| CHF | 2.1 | 4.0 | <0.001 |
| CAD | 7.8 | 9.9 | <0.001 |
| Neurological disorders | 1.2 | 2.2 | <0.001 |
| Coagulopathy | 2.0 | 3.4 | 0.10 |
| Metastatic Cancer | 17.4 | 21.9 | <0.001 |
| Diabetes | 15.5 | 19.0 | <0.001 |
| Hypertension | 42.0 | 48.3 | <0.001 |
| Chronic Lung Disease | 9.6 | 14.3 | <0.05 |
| Chronic Kidney Disease | 0.1 | 0.1 | 0.99 |
| History of Radiation | 33.3 | 39.9 | <0.001 |

Early: Discharge within 3 days of proctectomy; Routine: Discharge after 3 days of proctectomy; CHF: Congestive Heart Failure; CAD: Coronary Artery Disease; HTN: Hypertension.

## Factors associated with early discharge

Following adjustment, receiving care at a high volume center (AOR 1.30, 95% CI 1.13, 1.50), and laparoscopic approach (AOR 2.27, 95% CI 2.09, 2.47) were associated with increased likelihood of early discharge. Conversely, higher Elixhauser Comorbidity Index (AOR 0.83, 95% 0.80, 0.85), Medicaid insurance (AOR 0.62, 95% CI 0.55, 0.70), metastatic cancer (AOR 0.87, 95% CI 0.78, 0.96; Fig 2), and history of radiation (AOR 0.75, 95% CI 0.69, 0.82 Table 2); were associated with decreased odds of early discharge. Ileostomy creation did not alter odds of early discharge (AOR 1.03; 95% CI 0.95, 1.12; Table 2).

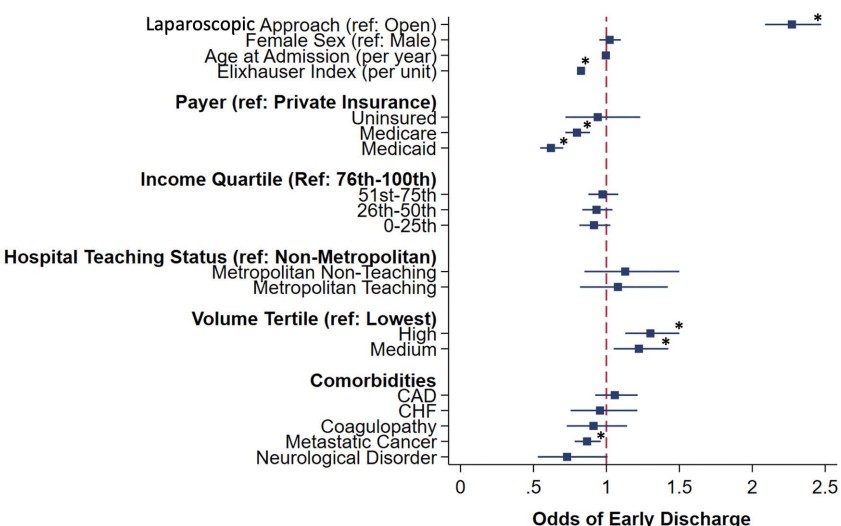

**Fig 2. Factors associated with early discharge.** *CHF*: Congestive Heart Failure; *CAD*: Coronary Artery Disease.

**Table 2. Adjusted odds of early discharge.**

|  | AOR | 95% CI | p-value |
|---|---|---|---|
| Hypertension | 1.3 | 1.18, 1.43 | <0.001 |
| Diabetes | 1.21 | 1.09, 1.35 | <0.001 |
| Chronic Lung Disease | 1.19 | 1.04, 1.36 | <0.05 |
| Chronic Kidney Disease | 0.46 | 0.17, 1.25 | 0.13 |
| Ileostomy Creation | 1.03 | 0.95, 1.12 | 0.45 |
| CHF | 1.09 | 0.85, 1.38 | 0.50 |
| CAD | 1.02 | 0.89, 1.17 | 0.76 |
| Coagulopathy | 1.06 | 0.84, 1.33 | 0.64 |
| Metastatic Cancer | 0.87 | 0.78, 0.96 | <0.05 |
| History of Radiation | 0.75 | 0.69, 0.82 | <0.001 |

CI: Confidence Interval; AOR: Adjusted Odds Ratio

## Clinical and financial outcomes Fillowing early discharge

Early discharge on the index hospitalization was associated with reduced rates of non-elective readmission within 30 days (11.7 vs 15.6%, p<0.001; Table 3). Additionally, *Early* less frequently developed infectious (21.5 vs 27.5%, p<0.01), or anastomotic leak complications (1.0 vs 4.9%, p<0.001, Table 3) during non-elective readmission. There was no significant difference in the rates of cardiac, respiratory, or renal complications between the two cohorts. Moreover, *Early* had lower 30-day cumulative LOS (3 [3–3] vs 6 days [4 –8], p<0.001) and hospitalization costs ($33,000 [25,000–45,000] vs. 39,000 [29,000–54,000], p<0.001) upon readmission. Notably, in-hospital mortality during readmission was insignificant between the two cohorts (0.7 vs 0.9%, p=0.65).

Following adjustment with entropy balancing and achieving acceptable covariate balance (Fig 3), early discharge remained independently associated with reduced odds of non-elective readmission within 30 days (AOR 0.77, 95% CI 0.69, 0.86; Fig 4; Table 4). Additionally, following non-elective readmission, *Early* less frequently developed infectious complications (AOR 0.77, 95% CI 0.65, 0.92). Moreover, early discharge was associated with a reduced 30-day cumulative LOS (β −2.8, 95% CI −2.9, −2.7) and hospitalization costs (β -$6,200, 95% CI $-8,600, −3,700). There remained no significant difference in odds of cardiac, respiratory, nor renal complications (Table 4). Early discharge did not alter in-hospital mortality during readmission (AOR 0.69, 95% CI 0.33, 1.45, p=0.32). In a subgroup analysis restricted to patients without complications during the index admission, early discharge remained independently associated with reduced odds of non-elective readmission within 30 days (AOR 0.79, 95% CI 0.70, 0.88), less frequent infectious complications (AOR 0.80, 95% CI 0.67, 0.96), and reduced cumulative 30-day length of stay (β −2.7, 95% CI −2.8, −2.6) and hospitalization costs (β -$5,200, 95% CI $-7,800, −2,500) (Table 5).

## Discussion

The present study examined the association of early discharge with clinical and financial outcomes following proctectomy for rectal cancer. We found that 1 in 4 patients were discharged within 3 days of proctectomy. Patients with expedited discharge had a lower Elixhauser Comorbidity Index, more commonly received a laparoscopic proctectomy, and were more frequently treated at high volume centers. Additionally, *Early* had reduced 30-day cumulative LOS, risk of non-elective readmission, and hospitalization costs. Importantly, we noted that early discharge did not alter in-hospital mortality during readmission, suggesting that expedited discharge does not compromise patient safety. Several of our findings warrant further discussion.

**Table 3. Unadjusted outcomes for proctectomy patients upon readmission stratified by early discharge.**

| | Early discharge | Routine Discharge | p-value |
|---|---|---|---|
| 30 Day Non-elective Readmission (%) | 11.7 | 15.6 | <0.001 |
| In-hospital mortality (%) | 0.7 | 0.9 | 0.65 |
| *Major Complications (%)* | | | |
| Cardiac | 3.0 | 1.9 | 0.11 |
| Respiratory | 2.1 | 3.3 | 0.13 |
| Infectious | 21.5 | 27.5 | <0.01 |
| Renal | 24.3 | 24.4 | 0.97 |
| Anastomotic Leak | 1.0 | 4.9 | <0.001 |
| Any complications | 44.8 | 52.1 | <0.005 |
| *Cumulative 30 Day Resource Use* | | | |
| Costs ($) | 33,000 [25000 - 45000] | 39,000 [29000 - 54000] | <0.001 |
| LOS (d) | 3 [3 – 3] | 6 [4 – 8] | <0.001 |

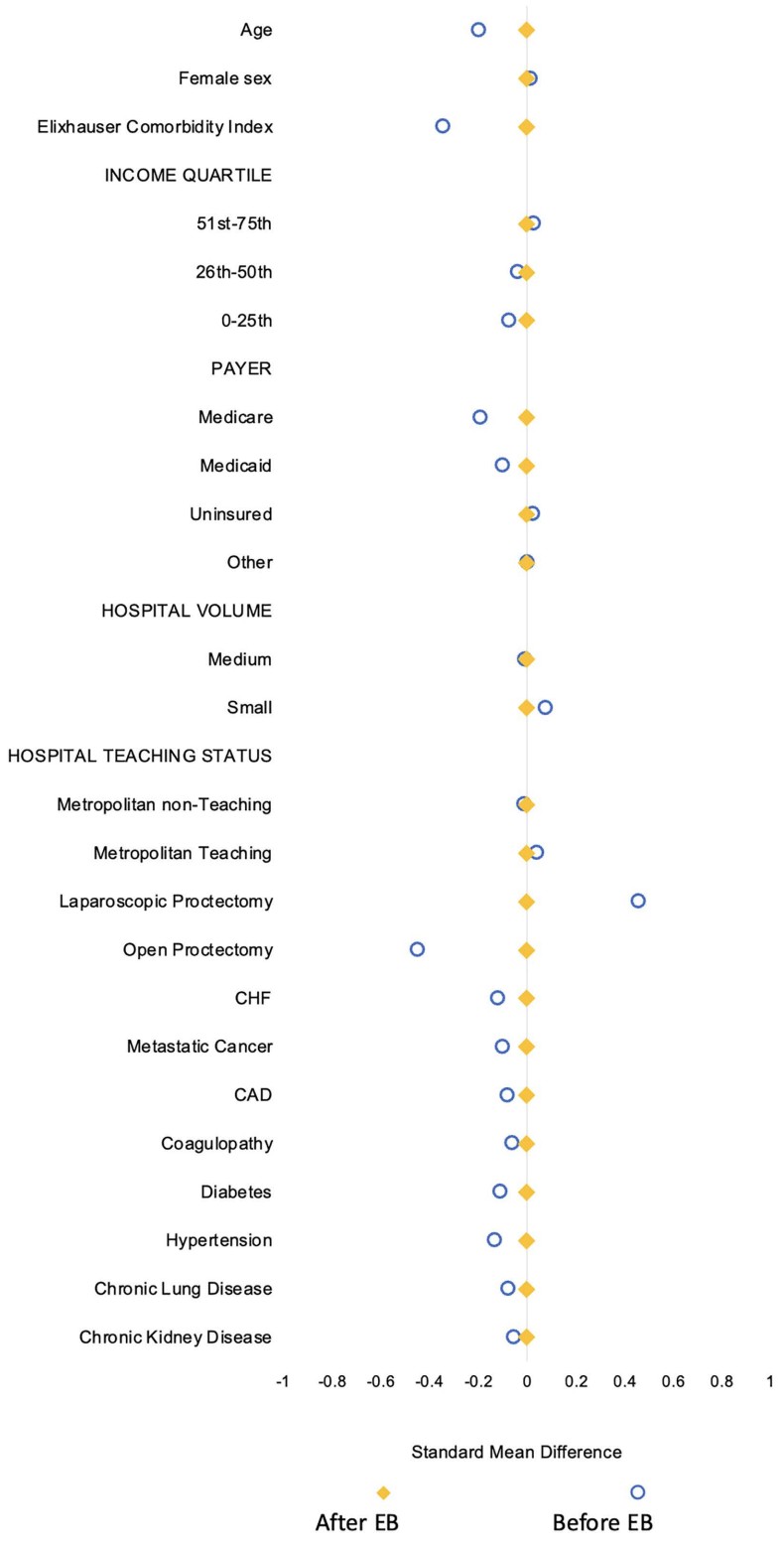

**Fig 3. Pre- and post- covariate balancing after entropy balancing.** EB: entropy balancing.

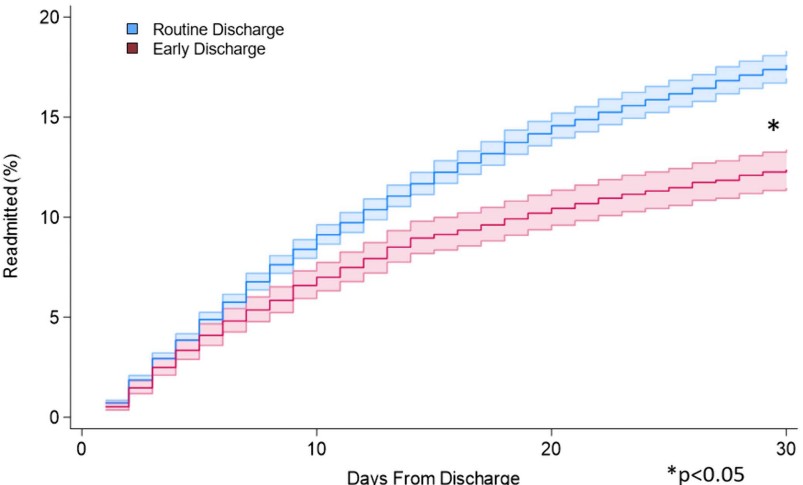

**Fig 4. Proportion of patients readmitted within 30 days of index discharge stratified by Early and Routine following proctectomy.** *Early: Patients discharged within 3 days of proctectomy; Routine: Patients discharged after 3 days of proctectomy.*

**Table 4. Entropy balanced risk-adjusted outcomes for proctectomy patients upon readmission stratified by early discharge.**

| Risk-adjusted | Early Discharge | 95% CI | p-value |
|---|---|---|---|
| Non-elective Readmission, AOR | 0.77 | 0.69, 0.86 | <0.001 |
| In-hospital mortality, AOR | 0.69 | 0.33, 1.45 | 0.32 |
| *Major Complications, AOR* | | | |
| Cardiac | 1.37 | 0.88, 2.14 | 0.16 |
| Respiratory | 0.81 | 0.56, 1.19 | 0.29 |
| Infectious | 0.77 | 0.65, 0.92 | <0.01 |
| Renal | 0.97 | 0.82, 1.14 | 0.68 |
| Any Complications | 0.81 | 0.71, 0.91 | <0.01 |
| *30 day Resource use (β)* | | | |
| Costs ($1,000) | −6.2 | −8.6, −3.7 | <0.001 |
| LOS (d) | −2.8 | −2.9, −2.7 | <0.001 |

CI: Confidence Interval; AOR: Adjusted Odds Ratio

In recent years, early discharge following proctectomy has become increasingly common and has been linked with reduced resource utilization and improved outcomes, similar to the present study [8,21]. Turrentine and colleagues found that early discharge was associated with lower rates of unplanned readmission after proctectomy [8]. Moreover, recent advances in minimally invasive surgery and the advent of Enhanced Recovery After Surgery (ERAS) protocols have supported early discharge following colorectal resection [22]. While proctectomy-specific adoption of ERAS pathways has been sparse in the literature, our study supports its feasibility when clinically appropriate. We found that while age was comparable between the two cohorts, *Early* had lower burden of comorbidities as measured by the Elixhauser Comorbidity Index. This association between early discharge and reduced burden of comorbidities has been reported across a variety of other procedures including esophagectomy and pancreatoduodenectomy [23,24]. Efforts to advance early discharge have been particularly prominent in elective colectomy. Recent national studies now demonstrate the feasibility and cost-effectiveness of ambulatory surgery for elective minimally invasive colectomies [25]. Unfortunately, our

**Table 5. Risk-adjusted outcomes after proctectomy for patients with no complication upon readmission stratified by early discharge.**

|  | Early discharge | 95% CI | p-value |
|---|---|---|---|
| 30 Day Non-elective Readmission (%) | 0.79 | 0.70–0.88 | <0.001 |
| In-hospital mortality (%) | 0.71 | 0.32–1.57 | 0.4 |
| *Major Complications (%)* |  |  |  |
| Cardiac | 1.25 | 0.77–2.03 | 0.89 |
| Respiratory | 0.79 | .53–1.17 | 0.24 |
| Infectious | 0.80 | 0.67–0.96 | <0.05 |
| Renal | 0.96 | 0.81–1.14 | 0.64 |
| Any complications | 0.84 | 0.74–0.96 | <0.01 |
| *Cumulative 30 Day Resource Use* |  |  |  |
| Costs ($1,000) | −5.2 | −7.8 - −2.5 | <0.001 |
| LOS (d) | −2.7 | −2.8 - −2.6 | <0.001 |

results showed that Medicaid and Medicare insurances were associated with decreased odds of early discharge, underscoring the need for continued efforts to optimize care for vulnerable patient populations. Future studies should aim to further identify patient characteristics that would be most amenable to early discharge in rectal cancer patients following proctectomy.

In the present work we found that high volume centers were associated with increased likelihood of early discharge. This finding is consistent with the robust body of literature describing a positive relationship between surgical volume and early discharge [26,27]. This relationship is not unique to proctectomy and has been attributed to the specialized expertise, streamlining of surgical care, and ready adoption of standardized protocols that is often present at higher throughput centers [28–30]. Moreover, high volume hospitals may be more likely to have well-established multidisciplinary teams that ensure comprehensive perioperative care, reducing complications and optimizing recovery time without compromising care quality [31]. It is for this reason that over the last decade, centralization for rectal cancer care has been increasing. Choi and colleagues found a positive volume outcome relationship among patients undergoing proctectomy. However, the authors also cautioned that centralization could widen inequalities in access to healthcare especially in rural, low socioeconomic, and older patient populations that may experience significant transportation barriers to accessing centers of excellence [32]. Therefore, the benefits of centralizing care at high-volume centers must be carefully balanced against the transportation challenges patients may face in accessing these facilities.

Following entropy balanced risk-adjustment, *Early* accrued significantly reduced 30-day cumulative hospitalization costs and non-elective readmission. This finding corroborates with previous studies that found the implementation of early discharge following colorectal surgery to be effective in reducing healthcare costs [33–35]. This reduction in hospitalization costs has been linked to more streamlined care and reduction in the daily costs of inpatient hospitalization [33]. The financial benefit of early discharge has been two-fold. It's been proposed that while the reduced cost significantly lessens the financial burden of proctectomy, the major financial impact of early discharge is in the added revenue from the increased capacity for additional hospital admissions [33]. Critics of early discharge point to a subset of patients who experience increased total hospitalization costs due to bounce-back readmissions, defined as readmission within seven days of the initial discharge [35]. However, the findings of this study show that the total 30-day hospitalization costs, including both the initial stay and any readmissions, were still lower in the early discharge group.

This study must be interpreted in light of several limitations. Because we conducted a retrospective review of an administrative database, our findings are observational and cannot determine causality. Furthermore, the NRD lacks detailed clinical information that would help contextualize patient selection for proctectomy such as laboratory profiles, imaging findings, tumor stage, or anatomic complexity which limits our ability to account for oncologic or disease-specific

factors. Additionally, operational details surrounding perioperative management are also unavailable. Institutional protocols, surgeon-level decision patterns, and center-specific approaches to postoperative care may differ substantially across hospitals but cannot be examined within this dataset. Likewise, information about whether patients were managed under enhanced recovery pathways or other structured discharge processes is not included, preventing us from assessing the influence of standardized fast-track strategies on discharge timing. Specifically, patient selection criteria for early discharge such as surgeon preference, patient functional status are not captured in the NRD and could not be accounted for. These unmeasured factors represent an important source of potential selection bias. Relatedly, although we restricted our cohort to patients with a diagnosis of rectal cancer, we did not distinguish cases performed for concurrent benign indications or concurrent proctocolectomy, which may introduce additional heterogeneity in surgical complexity and postoperative recovery. Although ICD-10 coding allows for identification of specific metastatic sites, site-specific stratification was not performed in the present study, as our primary aim was to account for the presence of metastatic disease as a marker of systemic disease severity. Additionally, the data source does not provide information about extraneous factors that may influence the likelihood of early discharge, including patients' specific support systems outside the hospital. Although the NRD allows tracking of readmissions and mortality, events occurring at non-NRD hospitals or deaths outside the inpatient setting are not captured. As a result, some post-discharge complications or outcomes may be underestimated. Finally, it is important to note that the reported resource utilization reflects the costs of admissions but does not account for the costs associated with post-discharge care and monitoring—an essential component of ERAS programs. Despite these limitations, we used a large, nationwide sample and statistical analyses to assess early discharge and clinical outcomes following proctectomy across the United States.

The present study used robust statistical methods to demonstrate reduced readmissions and hospitalization costs within 30 days of discharge in those discharged within 3 days of proctectomy for rectal cancer. We found several patient- and hospital-level factors that significantly alter rates of early discharge. Our findings suggest that early discharge is safe and cost-effective in rectal cancer patients when clinically appropriate. Further work is necessary to increase its adoption beyond high volume hospitals and privately insured patients.

## Author contributions

**Conceptualization:** Aboubacar Cherif, Troy N. Coaston, Saad Mallick, Zihan Gao, Esteban Aguayo, Peyman Benharash, Hanjoo Lee.

**Data curation:** Peyman Benharash.

**Formal analysis:** Aboubacar Cherif, Troy N. Coaston, Melissa Justo, Dariush Yalzadeh, Saad Mallick, Zihan Gao, Esteban Aguayo, Peyman Benharash, Hanjoo Lee.

**Investigation:** Aboubacar Cherif, Saad Mallick, Peyman Benharash, Hanjoo Lee.

**Methodology:** Aboubacar Cherif, Troy N. Coaston, Zihan Gao, Peyman Benharash, Hanjoo Lee.

**Project administration:** Aboubacar Cherif, Troy N. Coaston, Dariush Yalzadeh, Saad Mallick, Esteban Aguayo, Peyman Benharash, Hanjoo Lee.

**Resources:** Peyman Benharash.

**Software:** Peyman Benharash.

**Supervision:** Troy N. Coaston, Saad Mallick, Esteban Aguayo, Peyman Benharash, Hanjoo Lee.

**Validation:** Peyman Benharash, Hanjoo Lee.

**Visualization:** Aboubacar Cherif, Zihan Gao, Esteban Aguayo, Peyman Benharash, Hanjoo Lee.

**Writing – original draft:** Aboubacar Cherif, Melissa Justo, Dariush Yalzadeh, Saad Mallick, Peyman Benharash, Hanjoo Lee.

**Writing – review & editing:** Aboubacar Cherif, Troy N. Coaston, Melissa Justo, Dariush Yalzadeh, Saad Mallick, Zihan Gao, Esteban Aguayo, Peyman Benharash, Hanjoo Lee.

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
