## [Decision Letter · Decision Letter 0]

21 Jun 2025

PONE-D-25-24304Association of Early Discharge and Clinical Outcomes Following ProctectomyPLOS ONE

Dear Dr. Lee,

Thank you for submitting your manuscript to PLOS ONE. After careful consideration, we feel that it has merit but does not fully meet PLOS ONE’s publication criteria as it currently stands. Therefore, we invite you to submit a revised version of the manuscript that addresses the points raised during the review process.

If applicable, we recommend that you deposit your laboratory protocols in protocols.io to enhance the reproducibility of your results. Protocols.io assigns your protocol its own identifier (DOI) so that it can be cited independently in the future. For instructions see: https://journals.plos.org/plosone/s/submission-guidelines#loc-laboratory-protocols. Additionally, PLOS ONE offers an option for publishing peer-reviewed Lab Protocol articles, which describe protocols hosted on protocols.io. Read more information on sharing protocols at . Additionally, PLOS ONE offers an option for publishing peer-reviewed Lab Protocol articles, which describe protocols hosted on protocols.io. Read more information on sharing protocols at https://plos.org/protocols?utm_medium=editorial-email&utm_source=authorletters&utm_campaign=protocols..

We look forward to receiving your revised manuscript.

Kind regards,

Kuo-Cherh Huang

Academic Editor

PLOS ONE

Journal Requirements:

https://pubmed.ncbi.nlm.nih.gov/38938260/

https://www.clinicalkey.com/#!/content/playContent/1-s2.0-S0003497521007013?returnurl=https:%2F%2Flinkinghub.elsevier.com%2Fretrieve%2Fpii%2FS0003497521007013%3Fshowall%3Dtrue&referrer=https:%2F%2Fapi.ithenticate.com%2F

In your revision ensure you cite all your sources (including your own works), and quote or rephrase any duplicated text outside the methods section. Further consideration is dependent on these concerns being addressed

4. Please ensure that you refer to Figure 4 in your text as, if accepted, production will need this reference to link the reader to the figure.

5. Please remove all personal information, ensure that the data shared are in accordance with participant consent, and re-upload a fully anonymized data set.

Additional Editor Comments :

Dear Dr, Lee,

Thank you for your submission to PLoS ONE. I have received the feedback from two experts in the field of surgery, and the reviewers had provided thorough reviews and specific comments on your manuscript, especially in the Methods and Results sections. Please respond to each comment of the reviewers carefully and thoroughly. Please explain where you feel you cannot completely agree with reviewers’ suggestions. Thank you.

Kuo-Cherh Huang

Academic Editor

Reviewers' comments:

Reviewer's Responses to Questions

**Comments to the Author**

1. Is the manuscript technically sound, and do the data support the conclusions?

Reviewer #1: Partly

Reviewer #2: Partly

2. Has the statistical analysis been performed appropriately and rigorously? 

Reviewer #1: Yes

Reviewer #2: I Don't Know

3. Have the authors made all data underlying the findings in their manuscript fully available?

Reviewer #1: Yes

Reviewer #2: Yes

4. Is the manuscript presented in an intelligible fashion and written in standard English?

Reviewer #1: Yes

Reviewer #2: Yes

5. Review Comments to the Author

Reviewer #1: Aboubacar and colleagues used a nationally-representative dataset and evaluated expedited discharge after proctectomy (defined as =<3 days from prior literature). They hypothesized that patients discharged early would have lower readmission and resource use. They found significant cohort differences among the groups. Patients discharged early were found to have lower readmission and resource utilization, which was consistent with their hypothesis. The authors conclude that early discharge is feasible in the right patient population.

The statistics are clear and the paper is generally well-written; however, there are many methodological issues as detailed below. It is clear that the patient groups were inherently not similar. There is also significant selection bias in those who were discharged early. I have the following questions, concerns, and recommendations:

Major:

- Most critically, it’s not surprising that healthier patients with less extensive disease and comorbidities would be better candidates for early discharge. If the cohorts were truly balanced (comparing similar patient populations), one could draw stronger conclusions from this paper. As currently presented, being discharged early seems to be driven more by an inherently healthier patient than by physician choice to “push the envelope” and expedite a patient’s discharge. The paper would be more useful if framed from the latter. If the patient cohorts were better matched, I would expect that readmissions and complication rates to be similar, not improved

- Given the large proportion of laparoscopic proctectomy in the Early group, this would suggest there was clear selection bias that the study may not have granularity to ascertain. I would suggest that the authors perform a secondary/sensitivity analysis either using a propensity-matched analysis (such as inverse probability of treatment weighing or entropy balancing) and/or perform a separate analysis evaluating only patients who underwent laparoscopic surgery

- Presence of metastatic disease, which was higher in the Routine group, would also suggest that the groups were not balanced and Routine patients had more complex burden of disease, including radiation

- Given the inability to know the timing of diagnoses, how are the authors able to assess whether a complication prevented a patient from being discharged Early? It looks like the authors looked at complications at the readmission, but didn’t make any comparisons during the index hospitalization. This would obviously skew who would be Early

- How did the authors define a “major complication?” As above, depending on what it is, it may delay a discharge and cause selection bias. The authors should consider performing separate analysis removing patients who underwent major complications during the index hospitalization

- It is also possible that a complication could cause a significant increase in the LOS. I suggest that the authors omit patients who stayed significantly longer than expected. This should improve the homogeneity of their cohort

- What were the reasons for readmissions for those requiring readmission? The non-elective readmission rate for both groups is relatively high. I’m not sure about the clinical significance of reporting mortality on the readmission given how low this value is. Discussing needing blood products upon readmission may not be a clinically helpful metric either. Critically, did any of them develop leaks and how did this compare between groups? This would be a more clinically-relevant reason for readmission to report and would be more important for the readership

- Are the authors able to specify if a patient received a LAR or APR? What about how many ended up getting ileostomies? Obviously extent of operation would determine who could be a candidate for early discharge

- The authors should mention more clearly in their limitations that patient selection criteria is unable to be accounted for. This is the largest limitation of the study and should be mentioned explicitly

Minor:

- What proportion of patients were robotic-assisted?

- Are the authors able to specify where the rectal cancer metastasis was located? Were there patients that may have had concomitant liver resection?

- How many patients had a history of radiation?

- How did the authors calculate cumulative LOS in the results section? It would be obvious that the LOS in Early would be shorter than that of Routine in the index hospitalization. This should be better defined in the methods section

- Why do the authors think that early patients had more cardiac complications? This is mentioned in the results and not addressed afterwards

- The authors should further discuss the discrepancy of Turrentine study vs their own as opposed to just acknowledging this paper as contrary to their findings

- The authors selected only a few co-morbidities in Table 1. I would encourage the authors to include and adjust for additional co-morbidities such as diabetes, hypertension, chronic lung disease, chronic kidney disease. It’s not clear why the authors chose to report neurologic disorders which should not be clinically-relevant in this patient population

- In Table 1, the authors should avoid casual language such as “lap” or “comp”

- Although mentioned in the figure, the text should also clearly state that patients with IBD were omitted

- Typo on figure 2 should read “laparoscopic”

Reviewer #2: Dear Editors,

Thank you for the opportunity to review this important study on early discharge following proctectomy. As authors point out, data on early discharge and enhanced recovery following proctectomy is not as robust when compared to those of colectomies, and the topic is an important and understudied area of research in colorectal surgery. The paper is generally well written and has potential to further inform generalizability and feasibility of early discharge following proctectomy. The authors are also well aware of the limitations of the study, and outline some in the discussion section, as well as need for further studies on the topic.

That said, the manuscript would benefit from addressing several key limitations, specifically in its methodology section. I recommend acceptance pending major revision. Below are specific suggestions organized by manuscript section:

Introduction

-The introduction is solid and highlights the gaps in existing literature. Although authors mention 3 days as the cutoff to define early discharge, most of the quoted literature pertain to colonic procedures or do not distinguish between colonic and rectal procedures. Given the more complex nature of proctectomies, the cutoff for early discharge after proctectomy is debated in the literature. The introduction could be further strengthened by explaining the rationale for choosing 3 days as cutoff to define early discharge specifically after proctectomy, quoting definitions and literature that are specific to proctectomy.

Methods

-There are several components of the STROBE checklist that are missing, including sample size calculation, plans to address potential confounding, how missing data was handled, etc.

-were diagnosis codes for rectal cancer used in defining the study cohort? If so, what diagnosis codes were used? They should be listed, similar to how proctectomy codes were listed. From the flow diagram, it seems like UC or Crohn’s diagnosis was an exclusion factor. Did you also exclude patients who underwent proctectomy for other diagnoses other than cancer like rectourethral/vaginal fistula, proctitis, rectal trauma, rectal prolapse, FAP etc.? And did you exclude patients who underwent concurrent colectomy (i.e. proctocolectomy for IBD, FAP). More detail re: how the study cohort was defined is needed.

-the authors mention that elastic net regularization method to select model covariates. Did the authors also add clinically important covariates that were selected a priori, in addition to statistically selected covariates? Patients who are discharged early likely have several key underlying differences in patient, disease, and treatment-associated variables that need to be considered

-

Results

-The flow diagram needs more detail on how patients were included and excluded according to diagnosis codes, as outlined above

-presence of stoma (whether temporary diverting loop ileostomy or colostomy) is a major factor to consider when thinking about early discharge after proctectomy. Patients with new stoma need additional teaching and closer monitoring prior to discharge – was there any effort made to define which patients had concurrent stoma creation? This is an important variable to capture and report, as well as include in the statistical model

-the authors should describe which clinical factors were adjusted for in the final analysis

-what about anastomotic leak? Was this captured in the database? There are diagnosis codes for anastomotic leaks that have been used in previous studies using the NRD database – as this is a major complication specific to proctectomy, effort should be made to capture and report this in the most accurate way possible.

Discussion

-the discussion is well written, with detailed description of strengths and weaknesses of the paper, as well as future directions and relevance of the paper in clinical setting.

6. PLOS authors have the option to publish the peer review history of their article (what does this mean?). If published, this will include your full peer review and any attached files.). If published, this will include your full peer review and any attached files.

.

Reviewer #1: No

Reviewer #2: No

While revising your submission, please upload your figure files to the Preflight Analysis and Conversion Engine (PACE) digital diagnostic tool, https://pacev2.apexcovantage.com/. PACE helps ensure that figures meet PLOS requirements. To use PACE, you must first register as a user. Registration is free. Then, login and navigate to the UPLOAD tab, where you will find detailed instructions on how to use the tool. If you encounter any issues or have any questions when using PACE, please email PLOS at . PACE helps ensure that figures meet PLOS requirements. To use PACE, you must first register as a user. Registration is free. Then, login and navigate to the UPLOAD tab, where you will find detailed instructions on how to use the tool. If you encounter any issues or have any questions when using PACE, please email PLOS at figures@plos.org. Please note that Supporting Information files do not need this step.. Please note that Supporting Information files do not need this step.

---

## [Author Response · Author response to Decision Letter 1]

12 Dec 2025

Reviewer #1:

Major:

1a. Most critically, it’s not surprising that healthier patients with less extensive disease and comorbidities would be better candidates for early discharge. If the cohorts were truly balanced (comparing similar patient populations), one could draw stronger conclusions from this paper. As currently presented, being discharged early seems to be driven more by an inherently healthier patient than by physician choice to “push the envelope” and expedite a patient’s discharge. The paper would be more useful if framed from the latter. If the patient cohorts were better matched, I would expect that readmissions and complication rates to be similar, not improved.

1b. Given the large proportion of laparoscopic proctectomy in the Early group, this would suggest there was clear selection bias that the study may not have granularity to ascertain. I would suggest that the authors perform a secondary/sensitivity analysis either using a propensity-matched analysis (such as inverse probability of treatment weighing or entropy balancing) and/or perform a separate analysis evaluating only patients who underwent laparoscopic surgery

1c. Presence of metastatic disease, which was higher in the Routine group, would also suggest that the groups were not balanced and Routine patients had more complex burden of disease, including radiation

We thank the reviewer for these important comments. We fully agree that the baseline imbalance and selection bias are inherent limitations in retrospective database studies of discharge timing. We have addressed your suggestions and performed entropy balancing to construct a weighted comparison group with closely matched covariate distributions across patient and hospital characteristics. This method has been shown to outperform traditional propensity score matching by improving covariate balance without reducing sample size. After entropy balancing, all standardized mean differences were <0.1 (Fig 3), indicating excellent balance of surgical approach, comorbidity burden, and metastatic disease. Importantly, the associations between early discharge and decreased readmissions, LOS, and hospitalization costs remained statistically significant after weighting, suggesting that findings were not solely driven by healthier baseline profiles.

2. Given the inability to know the timing of diagnoses, how are the authors able to assess whether a complication prevented a patient from being discharged Early? It looks like the authors looked at complications at the readmission, but didn’t make any comparisons during the index hospitalization. This would obviously skew who would be Early

We thank the reviewer for highlighting this concern. The NRD does not provide time-stamped intra-hospital complication data (i.e whether complications occurred before or after operation). For this reason, we could not accurately use complications during index admission to determine impact on discharge after proctectomy.

3a. How did the authors define a “major complication?” As above, depending on what it is, it may delay a discharge and cause selection bias. The authors should consider performing separate analysis removing patients who underwent major complications during the index hospitalization

3b. It is also possible that a complication could cause a significant increase in the LOS. I suggest that the authors omit patients who stayed significantly longer than expected. This should improve the homogeneity of their cohort

We thank the reviewer for this suggestion. In the NRD, complications are captured using ICD-10 codes and grouped into validated categories (cardiac, respiratory, infectious, renal, thromboembolic, and transfusion-related). In our study, major complications included: cardiac, respiratory, infectious, renal, thromboembolic, transfusion-related, and strokes. However, the NRD does not provide timestamped information for complications occurring during index hospitalization which prevents us from reliably excluding patients whose early postoperative complication may have directly impacted their discharge timing. But because severe postoperative complications often do result in prolonged and clinically atypical lengths of stay, we excluded patients with an index LOS > 9 days which corresponded to the 90th percentile to decrease the heterogeneity of the cohort.

“Finally, patients with a length of stay greater than 9 days (90th percentile for length of stay) were excluded to decrease the heterogeneity of the cohort.”

4. What were the reasons for readmissions for those requiring readmission? The non-elective readmission rate for both groups is relatively high. I’m not sure about the clinical significance of reporting mortality on the readmission given how low this value is. Discussing needing blood products upon readmission may not be a clinically helpful metric either. Critically, did any of them develop leaks and how did this compare between groups? This would be a more clinically-relevant reason for readmission to report and would be more important for the readership

We thank the reviewer for this valuable suggestion. Unfortunately, the NRD does not provide granular clinical details regarding the specific cause of readmission (e.g anastomatic leak, abscess, etc.) which limits our ability to directly compare clinically meaningful etiologies such as leaks across early and routine discharge groups.

Regarding the readmission outcomes we reported, the non-elective readmission rate is indeed relatively high for both cohorts, which we believe reflects the overall complexity of proctectomy patients included in the NRD sample. Although readmission mortality was low, we reported it for completeness and transparency.

Similarly, the need for blood transfusion during readmission is available within the NRD as a coded variable but does not reliably distinguish between clinically meaningful etiologies and was included for completeness.

5. Are the authors able to specify if a patient received a LAR or APR? What about how many ended up getting ileostomies? Obviously extent of operation would determine who could be a candidate for early discharge

We appreciate this question. The NRD does not contain operative detail separating LAR from APR. However, ileostomy creation is captured and was incorporated into both the unadjusted and adjusted models. Ileostomy creation did not alter the likelihood of early discharge (AOR 1.03; 95% CI 0.95 , 1.12; Table 2). We have clarified this in Results.

6. The authors should mention more clearly in their limitations that patient selection criteria is unable to be accounted for. This is the largest limitation of the study and should be mentioned explicitly

We agree and have strengthened this statement in the limitations.

“Specifically, patient selection criteria for early discharge such as surgeon preference, patient functional status are not captured in the NRD and could not be accounted for. These unmeasured factors represent an important source of potential selection bias.”

Minor:

- What proportion of patients were robotic-assisted?

We thank the reviewer for this question. Robotic approach identifiers were not reliably available in the NRD because ICD-10-PCS differentiates between laparoscopic and open approaches but does not consistently capture robotic assistance.

- Are the authors able to specify where the rectal cancer metastasis was located? Were there patients that may have had concomitant liver resection?

We appreciate this point. The NRD does not specify the anatomic site of metastatic disease. Additionally, only 3 patients in the dataset had liver resections during index hospitalization.

- How many patients had a history of radiation?

About 34.6% of our cohort had prior radiation based on the NRD ICD-10 codes. However, these are rarely used and unreliable as it does not indicate location and further granular details regarding the radiation are not included.

- How did the authors calculate cumulative LOS in the results section? It would be obvious that the LOS in Early would be shorter than that of Routine in the index hospitalization. This should be better defined in the methods section

Thank you for the opportunity to clarify how 30-day cumulative LOS was calculated. Because the NRD captures all readmissions to participating hospitals within the same calendar year, we constructed a cumulative LOS variable by summing the index hospitalization LOS with any subsequent readmission occurring within 30 days of discharge. For patient with multiple readmissions, we summed the LOS of the index admission, the first readmission, and the LOS of a second readmission only when the second readmission also fell within the 30-day window.

We have clarified this in the methods.

“Cumulative length of stay (LOS) variable was constructed by summing the index hospitalization LOS with any subsequent readmission occurring within 30 days of discharge. For patient with multiple readmissions, we summed the LOS of the index admission, the first readmission, and the LOS of a second readmission only when the second readmission also fell within the 30 day window.”

- Why do the authors think that early patients had more cardiac complications? This is mentioned in the results and not addressed afterwards

We thank the review for this question. We believe the difference was most likely an artifact driven by very low event count. However, we found cardiac complications to no longer be significant after entropy balancing.

- The authors should further discuss the discrepancy of Turrentine study vs their own as opposed to just acknowledging this paper as contrary to their findings

Thank you for this comment. We appreciate the opportunity to clarify our comparison with the Turrentine study. We included it as an example of a study that corroborated our findings that early discharge was associated with lower rates of unplanned readmissions. Thus, there were no meaningful discrepancy between the two studies.

- The authors selected only a few co-morbidities in Table 1. I would encourage the authors to include and adjust for additional co-morbidities such as diabetes, hypertension, chronic lung disease, chronic kidney disease. It’s not clear why the authors chose to report neurologic disorders which should not be clinically-relevant in this patient population

We thank the reviewer and have updated Table 1 to include diabetes, hypertension, chronic lung disease, and CKD.

Neurological disorders were included in our comorbidity adjustment because, although they may not appear directly related to proctectomy, they meaningfully affect perioperative risk, functional status, and discharge disposition. Patients with underlying neurological conditions such as prior stroke, neuromuscular disease, or cognitive impairment may influence length of stay, timing of discharge, and likelihood of readmission, which are central outcomes in our study.

- In Table 1, the authors should avoid casual language such as “lap” or “comp”

All casual abbreviations have been revised to their full terminology throughout the manuscript.

- Although mentioned in the figure, the text should also clearly state that patients with IBD were omitted

We agree and added explicit text in the methods.

“Similarly, patients with a diagnosis of Ulcerative Colitis or Crohn’s disease were not included for analysis.”

- Typo on figure 2 should read “laparoscopic”

Thank you for this remark. The error has been corrected.

Reviewer #2:

Introduction

1a. The introduction is solid and highlights the gaps in existing literature. Although authors mention 3 days as the cutoff to define early discharge, most of the quoted literature pertain to colonic procedures or do not distinguish between colonic and rectal procedures. Given the more complex nature of proctectomies, the cutoff for early discharge after proctectomy is debated in the literature. The introduction could be further strengthened by explaining the rationale for choosing 3 days as cutoff to define early discharge specifically after proctectomy, quoting definitions and literature that are specific to proctectomy.

Thank you for this insightful comment. We agree that proctectomy is generally considered to be more complex than colectomy, and therefore the definition of “early discharge” warrants careful justification. Although much of the existing literature defines early discharge using a 3-day threshold in colorectal surgery broadly, several studies report that median LOS for proctectomy typically ranges from 5-6 days, which is comparable to the 4-7 days median LOS commonly reported for colectomy and supports the usage of a similar 3-day cutoff. We have clarified this rationale in the introduction and strengthened the justification with literature specific to proctectomy.

“Although proctectomy is generally considered to be more complex than colectomy, reported median lengths of stay for proctectomy (5-6 days) overlap with the 4-7 days commonly reported for colectomy, supporting the use of a similar 3-day threshold to identify patients discharged substantially earlier than contemporary practice patterns[8].”

Methods

-There are several components of the STROBE checklist that are missing, including sample size calculation, plans to address potential confounding, how missing data was handled, etc.

We appreciate this comment from the reviewer. A formal sample size calculation was not performed because our study utilized the NRD, a large administrative dataset with a fixed annual sample determined by the database design. The analytic cohort size reflects all eligible patients meeting inclusion criteria.

For our approach to confounding variables, we adjusted for a comprehensive set of demographic, comorbidity, procedural, and hospital-level variables. In addition, we used survey-weighted analyses to ensure national representativeness and correct variance estimation. Upon revision request, we also applied entropy balancing to further improve covariate balance between our cohorts.

For missing data, we clarified in the methods section that “missing key variables such as age, sex, and mortality were excluded.”

-Were diagnosis codes for rectal cancer used in defining the study cohort? If so, what diagnosis codes were used? They should be listed, similar to how proctectomy codes were listed. From the flow diagram, it seems like UC or Crohn’s diagnosis was an exclusion factor. Did you also exclude patients who underwent proctectomy for other diagnoses other than cancer like rectourethral/vaginal fistula, proctitis, rectal trauma, rectal prolapse, FAP etc.? And did you exclude patients who underwent concurrent colectomy (i.e. proctocolectomy for IBD, FAP). More detail re: how the study cohort was defined is needed.

We thank the reviewer for this important clarification request. We now explicitly list rectal cancer ICD-10 codes C20 which was used to identify patients with rectal cancer. Only patients with a primary diagnosis of rectal cancer who underwent proctectomy were included in the study cohort.

With respect to other indications, we did not specifically exclude patients who had concurrent diagnoses in addition to rectal cancer (i.e proctitis, fistula, rectal trauma, etc…) as long as rectal cancer was present. Our intention was to capture all patients undergoing proctectomy in the setting of rectal cancer, regardless of additional secondary diagnoses. Similarly, we did not exclude patients who underwent concurrent proctocolectomy if they also carried a rectal cancer diagnosis. We have clarified these points in the limitations.

“Although we restricted our cohort to patients with a diagnosis of rectal cancer, we did not distinguish cases performed for concurrent benign indications or concurrent proctocolectomy, which may introduce additional heterogeneity in surgical complexity and postoperative recovery.”

-The authors mention that elastic net regularization method to select model covariates. Did the authors also add clinically important covariates that were selected a priori, in addition to statistically selected covariates? Patients who are discharged early likely have several key underlying differences in patient,

---

## [Decision Letter · Decision Letter 1]

26 Jan 2026

PONE-D-25-24304R1Association of early discharge and clinical outcomes following proctectomy for patients with rectal cancer: a NRD analysisPLOS One

Dear Dr. Lee,

Thank you for submitting your manuscript to PLOS ONE. After careful consideration, we feel that it has merit but does not fully meet PLOS ONE’s publication criteria as it currently stands. Therefore, we invite you to submit a revised version of the manuscript that addresses the points raised during the review process.

If applicable, we recommend that you deposit your laboratory protocols in protocols.io to enhance the reproducibility of your results. Protocols.io assigns your protocol its own identifier (DOI) so that it can be cited independently in the future. For instructions see: https://journals.plos.org/plosone/s/submission-guidelines#loc-laboratory-protocols. Additionally, PLOS ONE offers an option for publishing peer-reviewed Lab Protocol articles, which describe protocols hosted on protocols.io. Read more information on sharing protocols at . Additionally, PLOS ONE offers an option for publishing peer-reviewed Lab Protocol articles, which describe protocols hosted on protocols.io. Read more information on sharing protocols at https://plos.org/protocols?utm_medium=editorial-email&utm_source=authorletters&utm_campaign=protocols..

We look forward to receiving your revised manuscript.

Kind regards,

Kuo-Cherh Huang

Academic Editor

PLOS One

Journal Requirements:

Additional Editor Comments:

Dear Dr. Lee,

Thank you for submitting your revised manuscript to PLoS ONE. Both experienced reviewers in the field of surgical oncology had carefully read your revised manuscript and responses to the prior round of review comments. Both referees appreciated your efforts. Having said that, they still had main concerns with your revised work. Yet, they also thoughtfully provided you with a number of concrete suggestions along with specific code references to help you further improve your study. Please respond to each comment of both reviewers carefully and thoroughly. Please explain where you feel you cannot completely agree with reviewer’s suggestions. Thank you.

Kuo-Cherh Huang

Academic Editor

Reviewer's Responses to Questions

**Comments to the Author**

1. If the authors have adequately addressed your comments raised in a previous round of review and you feel that this manuscript is now acceptable for publication, you may indicate that here to bypass the “Comments to the Author” section, enter your conflict of interest statement in the “Confidential to Editor” section, and submit your "Accept" recommendation.

Reviewer #1: (No Response)

Reviewer #2: All comments have been addressed

2. Is the manuscript technically sound, and do the data support the conclusions?

Reviewer #1: Yes

Reviewer #2: Yes

3. Has the statistical analysis been performed appropriately and rigorously? 

Reviewer #1: Yes

Reviewer #2: Yes

4. Have the authors made all data underlying the findings in their manuscript fully available?

Reviewer #1: Yes

Reviewer #2: Yes

5. Is the manuscript presented in an intelligible fashion and written in standard English?

Reviewer #1: Yes

Reviewer #2: Yes

6. Review Comments to the Author

Reviewer #1: I thank the authors for addressing some of my concerns. The manuscript is improved but requires additional revisions. I have several questions and/or concerns:

1. While it is true that timing of complication cannot be ascertained in their dataset, patients who had one or multiple complications may have still increased their LOS when they otherwise would not have if it was a routine operation. I strongly suggest that the authors perform a subgroup analysis omitting their complications of interest and seeing if this changes their results

2. There is a difference between completeness and clinical relevance. This data regarding readmission related death/transfusion should be moved to a supplemental table. If it’s not revisited by the authors in the discussion, it’s not worth including in the main body text

3. I disagree regarding the point of unable to assess location of metastatic disease. Please see: https://www.icd10data.com/ICD10CM/Codes/C00-D49/C76-C80/C78-/C78.7

4. Similarly, you should be able to differentiate APR vs LAR. Obviously APR involves a perineal resection that a LAR does not. I suggest the authors query there codes for these codes: https://cdn-links.lww.com/permalink/prs/d/prs_144_5_2019_09_26_kung_prs-d-18-02422_sdc1.pdf

5. The authors should also be able to evaluate for readmissions related to leaks: https://icdcodes.ai/diagnosis/anastomotic-leak/documentation

6. The robotic assistance codes are here: https://www.icd10data.com/ICD10PCS/Codes/8/E/0/W

7. History of radiation should be reported in the tables for completeness

Reviewer #2: Thank you to the authors for incorporating our feedback and addressing concerns.

We appreciate the explanations provided regarding sample size, missing data handling, diagnostic codes

Regarding Reviewer 1's request for differentiating LAR from APR - there are NRD codes that can allow for differentiation between the two, using ICD codes. There are also ICD codes available to characterize complications such as leak and abscess. Although as authors mentioned, there are no ICD codes exclusively specific for anastomotic leaks, there are codes available to infer a leak (i.e. Disruption of internal surgical wound, Peritoneal abscess, Fistula of intestine). You could use these in combination of procedure codes (i.e. re-laparotomy, lavage, percutaneous drainage etc.) to increase specificity for detecting the complication. This could be a helpful reference to address above issues (using NRD): https://link.springer.com/article/10.1007/s00464-021-08923-y

Otherwise, I believe the authors have addressed our concerns and the manuscript is improved from previous version.

7. PLOS authors have the option to publish the peer review history of their article (what does this mean?). If published, this will include your full peer review and any attached files.). If published, this will include your full peer review and any attached files.

.

Reviewer #1: No

Reviewer #2: No

---

## [Author Response · Author response to Decision Letter 2]

21 Mar 2026

Reviewer #1: I thank the authors for addressing some of my concerns. The manuscript is improved but requires additional revisions. I have several questions and/or concerns:

1. While it is true that timing of complication cannot be ascertained in their dataset, patients who had one or multiple complications may have still increased their LOS when they otherwise would not have if it was a routine operation. I strongly suggest that the authors perform a subgroup analysis omitting their complications of interest and seeing if this changes their results

We thank the reviewer for this insightful suggestion. To address the potential confounding effect of complications on LOS, we performed a subgroup analysis restricted to patients without complications during the index admission. Results are quoted below and reflected in the manuscript’s new version:

“subgroup analysis restricted to patients without complications during the index admission, early discharge remained independently associated with reduced odds of non-elective readmission within 30 days (AOR 0.79, 95% CI 0.70 , 0.88), less frequent infectious (AOR 0.80, 95% CI 0.67 , 0.96) and blood transfusion complications (AOR 0.68, 95% CI 0.47 , 0.99), and reduced cumulative 30-day length of stay (β -2.7, 95% CI -2.8 , -2.6) and hospitalization costs (β -$5,200, 95% CI $-7,800 , -2,500) (Table 5).”

2. There is a difference between completeness and clinical relevance. This data regarding readmission related death/transfusion should be moved to a supplemental table. If it’s not revisited by the authors in the discussion, it’s not worth including in the main body text

We appreciate this important distinction. In response, we have removed blood transfusion from the tables and result section as we don’t refer to it in the discussion. However, as the lack of increased mortality associated with early discharge is clinically important to our findings, we have succinctly referenced it in the discussion section as seen below:

“Importantly, we noted that early discharge did not alter in-hospital mortality during readmission, suggesting that expedited discharge does not compromise patient safety.”

3. I disagree regarding the point of unable to assess location of metastatic disease. Please see: https://www.icd10data.com/ICD10CM/Codes/C00-D49/C76-C80/C78-/C78.7

Thank you for this correction and for highlighting this. We have acknowledged these restrictions of our project in our methods and limitations as follows:

Methods:

“Metastatic disease was similarly identified using ICD-10-CM: C77, C78, C79, C80 and modeled as a composite variable to reflect the overall burden of advanced malignancy.”

Limitations:

“Although ICD-10 coding allows for identification of specific metastatic sites, site-specific stratification was not performed in the present study, as our primary aim was to account for the presence of metastatic disease as a marker of systemic disease severity.”

4. Similarly, you should be able to differentiate APR vs LAR. Obviously APR involves a perineal resection that a LAR does not. I suggest the authors query there codes for these codes: https://cdn-links.lww.com/permalink/prs/d/prs_144_5_2019_09_26_kung_prs-d-18-02422_sdc1.pdf

We thank the reviewer for this thoughtful suggestion. We carefully evaluated the use of ICD-10-PCS codes to differentiate APR from LAR.

In our study, we identified proctectomy using the ICD-10-PCS codes ODTP0ZZ (open) and ODTP4ZZ (laparoscopic), which have been previously validated and specifically correspond to abdominoperineal resection of the rectum. While additional ICD-10-PCS codes exist that may capture other rectal resections, including those suggested, several are categorized as “other” or represent procedures such as bypass operations which raises the concern for potential misclassification and heterogeneity. Because of this, we elected to use a more specific coding strategy to ensure cohort consistency.

5. The authors should also be able to evaluate for readmissions related to leaks: https://icdcodes.ai/diagnosis/anastomotic-leak/documentation

We appreciate this important suggestion. We have included leak-related complications during readmission as suggested. Please see cited results below:

“Additionally, Early less frequently developed infectious (21.5 vs 27.5%, p<0.01), or anastomotic leak complications (1.0 vs 4.9%, p<0.001, Table 3) during non-elective readmission.”

6. The robotic assistance codes are here: https://www.icd10data.com/ICD10PCS/Codes/8/E/0/W

We thank the review for this suggestion. We have now incorporated these codes to identify robot-assisted procedures. See result as cited below:

“Finally, Early was more likely to undergo a laparoscopic proctectomy (60.9 vs 39.0%, p<0.001) and an Ileostomy creation (43.7 vs 37.8%, p<0.001), but less likely to undergo robotic assistance (14.2 vs 25.7%, p<0.001) and less likely to have a history of radiation (33.3 vs 39.9%, p<0.001).”

7. History of radiation should be reported in the tables for completeness

We agree and have now included history of radiation therapy in the baseline characteristics table for completeness and transparency.

“Finally, Early was more likely to undergo a laparoscopic proctectomy (60.9 vs 39.0%, p<0.001) and an Ileostomy creation (43.7 vs 37.8%, p<0.001), but less likely to undergo robotic assistance (14.2 vs 25.7%, p<0.001) and less likely to have a history of radiation (33.3 vs 39.9%, p<0.001).”

Reviewer #2: Thank you to the authors for incorporating our feedback and addressing concerns.

We appreciate the explanations provided regarding sample size, missing data handling, diagnostic codes

Regarding Reviewer 1's request for differentiating LAR from APR - there are NRD codes that can allow for differentiation between the two, using ICD codes.

There are also ICD codes available to characterize complications such as leak and abscess. Although as authors mentioned, there are no ICD codes exclusively specific for anastomotic leaks, there are codes available to infer a leak (i.e. Disruption of internal surgical wound, Peritoneal abscess, Fistula of intestine). You could use these in combination of procedure codes (i.e. re-laparotomy, lavage, percutaneous drainage etc.) to increase specificity for detecting the complication. This could be a helpful reference to address above issues (using NRD): https://link.springer.com/article/10.1007/s00464-021-08923-y

We thank the reviewer for reinforcing these important methodological considerations and have addressed each one of them as seen above under reviewer 1.

We again thank the reviewers for their thoughtful and constructive feedback and are more than happy to revise it further as needed for it to be suitable for publication.

Much appreciated!

---

## [Decision Letter · Decision Letter 2]

12 Apr 2026

Association of early discharge and clinical outcomes following proctectomy for patients with rectal cancer: a NRD analysis

PONE-D-25-24304R2

Dear Dr. Lee,

We’re pleased to inform you that your manuscript has been judged scientifically suitable for publication and will be formally accepted for publication once it meets all outstanding technical requirements.

An invoice will be generated when your article is formally accepted. Please note, if your institution has a publishing partnership with PLOS and your article meets the relevant criteria, all or part of your publication costs will be covered. Please make sure your user information is up-to-date by logging into Editorial Manager at Editorial Manager® and clicking the ‘Update My Information' link at the top of the page. For questions related to billing, please contact  and clicking the ‘Update My Information' link at the top of the page. For questions related to billing, please contact billing support..

Kind regards,

Kuo-Cherh Huang

Academic Editor

PLOS One

Additional Editor Comments (optional):

Reviewers' comments:

Reviewer's Responses to Questions

**Comments to the Author**

1. If the authors have adequately addressed your comments raised in a previous round of review and you feel that this manuscript is now acceptable for publication, you may indicate that here to bypass the “Comments to the Author” section, enter your conflict of interest statement in the “Confidential to Editor” section, and submit your "Accept" recommendation.

Reviewer #1: All comments have been addressed

Reviewer #2: All comments have been addressed

2. Is the manuscript technically sound, and do the data support the conclusions?

Reviewer #1: Yes

Reviewer #2: Yes

3. Has the statistical analysis been performed appropriately and rigorously? 

Reviewer #1: Yes

Reviewer #2: Yes

4. Have the authors made all data underlying the findings in their manuscript fully available?

Reviewer #1: Yes

Reviewer #2: Yes

5. Is the manuscript presented in an intelligible fashion and written in standard English?

Reviewer #1: Yes

Reviewer #2: Yes

6. Review Comments to the Author

Reviewer #1: I thank the authors for adequately and succinctly addressing my concerns. Congratulations to them for an excellent paper

Reviewer #2: Comments have been addressed, thank you. I do not have any further request for edits or modifications.

7. PLOS authors have the option to publish the peer review history of their article (what does this mean?). If published, this will include your full peer review and any attached files.). If published, this will include your full peer review and any attached files.

.

Reviewer #1: No

Reviewer #2: No

---

## [Editor Report · Acceptance letter]

PONE-D-25-24304R2

PLOS One

Dear Dr. Lee,

I'm pleased to inform you that your manuscript has been deemed suitable for publication in PLOS One. Congratulations! Your manuscript is now being handed over to our production team.

Kind regards,

on behalf of

Dr. Kuo-Cherh Huang

Academic Editor

PLOS One